# Development of Polypropylene/Polyethylene Terephthalate Microfibrillar Composites Filament to Support Waste Management

**DOI:** 10.3390/polym13020233

**Published:** 2021-01-12

**Authors:** Abdulhakim Almajid, Rolf Walter, Tim Kroos, Harry Junaidi, Martin Gurka, Khalil Abdelrazek Khalil

**Affiliations:** 1Mechanical Engineering Department, College of Engineering, King Saud University, P.O. Box 800, Riyadh 11421, Saudi Arabia; hjunaidi@ksu.edu.sa; 2Engineering Management, College of Engineering, Prince Sultan University, P.O. Box 66833, Riyadh 11586, Saudi Arabia; 3Institute for Composite Materials (IVW GmbH), Technical University of Kaiserslautern, 67663 Kaiserslautern, Germany; rolf.walter@ivw.uni-kl.de (R.W.); Tim.Krooss@ivw.uni-kl.de (T.K.); martin.gurka@ivw.uni-kl.de (M.G.); 4Mechanical and Nuclear Engineering Department, College of Engineering, University of Sharjah, Sharjah 27272, UAE; kabdelmawgoud@sharjah.ac.ae

**Keywords:** PP/PET microfibrillar composites, waste management, recycling, automotive component

## Abstract

The concept of microfibrillar composites (MFCs) is adopted to produce composites of polyethylene terephthalate (PET) fiber-reinforced polypropylene (PP) materials. The two polymers were dry mixed with PET content ranging from 22 to 45 wt%. The PET has been used as a reinforcement to improve the mechanical properties of composites. The relationship between the morphology of the MFC structure and the mechanical behavior of the MFC filament was investigated. Analysis of the structure and mechanical behavior helped to understand the influence of the stretching ratio, extruder-melt temperature, stretching-chamber temperature, and filament speed.

## 1. Introduction

Because of the need for better sustainability, the prevention of waste, such as End of Life Vehicles (ELVs), is of significant concern for governmental regulations, (e.g., the ELV-directive of the European Union) [1]. Advanced materials and composites are used extensively in various parts in today’s vehicles. There are several factors that are leading research in this direction, including (i) improving quality of products, (ii) addressing environmental issues, (iii) developing a competitive economy, and (iv) addressing government regulations [2]. Developing low-weight and fuel-efficient vehicles with low emissions has directed research towards materials, design, and devices.

To address the ELV-directive of the European Union and the demand for energy consumption reduction, the automotive industry is focusing on producing lightweight materials based on the principle of polymer composites [3]. A light automobile can be achieved by reducing the product’s weight through the development of innovative structural designs and integrating applications of material and manufacturing technologies under the condition of cost control and performance improvement. Lightweight materials and components are crucial for the automobile industry to reduce emissions and to save energy. The reduction of an automobile’s deadweight by 10% could lower fuel consumption by 6–8% and could reduce emissions of CO_2_ by 14% [3]. The new electric automobiles (i.e., electro-cars) with their heavy batteries have even more urgent weight reduction requirements. The adaptation of polymers in the automotive industry to produce low-weight components is based on glass fiber-reinforced thermoplastics, which are mainly used to produce bumpers or other body-trim panels. Bumpers have functional and appearance requirements and should be able to withstand low-speed impacts and improve aerodynamic efficiency. Bumper covers are considered the most frequently replaced body part in cars [3].

Polypropylene (PP) and polyamide (PA) are the most commonly used thermoplastic matrices, while short or long glass fibers are used as a reinforcement. The recycling of composites that are composed of organic matrix and inorganic fibers is complicated. To mechanically recycle such a composite, it is generally milled, ground, cut, or shredded and then subsequently sorted to be used again as reinforcement or filler.

One potential solution for a more sustainable material are self-reinforcing thermoplastic polymer–polymer composites, like microfibrillar- or nanofibrillar-reinforced composites (MFCs) [4], which offer the possibility of recycling via a simple re-melting procedure [5,6,7]. Unlike fiber-reinforced composites, MFCs are considered polymer–polymer composites, where the polymer matrix is reinforced with polymer fibrils [8,9,10]. The processing–structure–property relationship for MFCs is an essential process in order to produce the desired filament [11,12]. To select the proper polymer pairs to produce an MFC, the following requirements should be met: (i) the polymers pairs should be thermodynamically immiscible; (ii) the difference in melting temperature between the polymers pairs should be at least 30 °K; and (iii) the polymer pairs should have good stretchability.

Estatiev and coworkers have developed the MFC concept on different length scales for microfibrillar- [4] and nanofibrillar-reinforced [5] materials. The MFC concept is based on the melt blending of two thermodynamically immiscible thermoplastic polymers. Their melting temperatures differ by at least 30 °K, followed by a subsequent stretching and annealing process, which leads to the thermoplastic composites being reinforced by the higher-melting polymer fibrils. The three significant steps in producing MFCs are:Melt blend two thermodynamically immiscible polymers whose melting temperatures differ by at least 30 °K; thenStretch the extruded blend at a temperature above the glass transition temperature of both polymers to orient the polymer phases (fibrillization); thenIsotropize the polymer matrix by an annealing process where the annealing temperature is above the matrix material’s melting temperature.

These three processing steps lead to micro-fibrillar polymer composites in which the micro-sized fibrils are embedded in a continuous matrix.

The MFCs created using this method have several advantages compared to fiber (macro-sized) reinforced composites, which are:Proper dispersion of the reinforcing fibrils is achieved as the fibrils are developed once the melt blend leaves the extruder, overcoming the problem that existed in fiber composites.As the fibril diameter is decreased and reaches a micro-scale, the critical length needed to create a fiber pullout is decreased, which reduces the fiber pullout problem.MFCs have a lower density compared to fiber-reinforced composites, which results in weight reduction.The MFC process is more environmentally friendly compared to fiber composites.The MFC process is fully recyclable.

It is worth noting here that the term miscibility is generally used interchangeably with compatibility. A thermodynamically miscible polymer does not separate into different phases and remains homogeneous down to the molecules [8,9]. A polymer blend with practical compatibility is merely a blend that shows useful commercial applications [13,14]. Thermodynamically immiscible polymers can be used in the process of fibrillation [15].

The primary goals of this investigation are, therefore, (i) to apply the experience gained on the MFC procedure for developing MFCs based on PET and PP to achieve improved mechanical and significantly impact properties that are appropriate for production of car bumpers, and (ii) to transfer the gained expertise into an industrial production process.

In these blends, PET fibrils play the role of reinforcing component, while the PP forms the matrix phase. The development process is divided into two steps: (i) the development of optimized MFC filaments with good mechanical properties, a high stretching ratio, and a microstructure leading to outstanding mechanical and crash properties that can be further developed as composites, and (ii) the development of a composite manufacturing process to form MFCs out of filaments or textile preforms. For both development steps, the relationship between the morphology and the mechanical and thermomechanical properties of test specimens with an MFC structure were characterized. Analysis of the structure and mechanical behavior helped to understand the stretching ratio’s influence, extruder melts temperature, stretching chamber temperature, and filament speed. Finally, it should be underlined that all proposed blends are entirely recyclable and can be re-used. In this paper, we present the MFC development process.

## 2. Materials and Methods

### 2.1. Materials

To develop the microfibrillar composite, PET was used as a filler in the PP matrix. The SABIC Company (Riyadh, Saudi Arabia) manufactured all materials. The manufacturer’s datasheet for the materials used in the MFC processing is presented in Table 1 as are the processing parameter used in producing the MFC.

### 2.2. Rheology

The rheology of the MFC base materials (Table 1) was characterized using a plate–plate rheometer. Disks of neat polymers (25 mm in diameter and 2 mm in thickness) were pressed at a temperature of 275 °C. The materials’ rheology was evaluated at 275 °C using an ARES parallel plate rheometer (Rheometric Scientific, East Windsor, NJ, USA) with a gap of 1 mm. All the measurements were performed in frequency mode with a strain amplitude of 10%.

### 2.3. Blending and Filament Processing

#### 2.3.1. Blending of MFC

The two polymers were dry-mixed at various PP/PET ratios by weight, with PET contents ranging from 22 to 45 wt%. The granules were melted, blended, and extruded in a twin-screw extruder (PL2000, Brabender GmbH & Co., KG Duisburg, Germany). Two melt-extrusion temperatures were examined at 255 °C and 269 °C. The diameter of the screws was 25 mm and the L/D ratio (length/diameter) was 22. The influence of the compound parameters on the quality of the blend was investigated by varying the extruder die diameter, extruder throughput rate, and melt temperature via extruder barrel heating. The objective is to get well-dispersed PET in the matrix with small particle size distribution and particle sizes in the range of 1–20 µm.

#### 2.3.2. Filament Stretching and Processing

Figure 1 shows a schematic view of the MFC process. Once the blend leaves the capillary die (diameter 2 mm), the molten extrudes are cooled down in a water bath to a temperature of approximately 50 °C. With a speed-controlled drive unit with initial speed *v_1_*, the extrudes are led into a hot chamber in which the temperature is kept above the glass temperature, *T**_g_*, of all the used polymers. They are continuously stretched (through neck formation). The stretching ratio is controlled using a second drive unit with speed *v_2_*. Finally, the stretched filaments are wound onto a spool. Using the design of the experiment approach, the stretching limits of PP/PET is determined. The objective is to achieve the maximum stable stretching ratio. The PP/PET filaments with a PET content of 22 wt% possess a diameter of about 0.5–0.9 mm after stretching, which corresponds to a stretch ratio of 8–15. The influence of the extruder settings (extruder speed, barrel temperatures) and the stretching line are investigated. Fundamental variations of the key parameters of the line are done with the system PP and 22 wt% PET. The following parameters variations were performed:The speed of the filament before entering the stretching chamber (6.5–12.5 cm/s in 3 steps)The temperature of the stretching chamber (90, 105, 120 °C)The melt temperature via extruder barrel heating (258, 269 °C)

The effect of the stretch ratio on the above processing parameter was investigated.

By a mechanical testing of the filaments, the best parameters were selected for manufacturing filaments with various PET contents. All filaments that were further investigated are processed with a 260 °C melt temperature, 20 rpm extruder speed, 2 mm die, 3 kg/h throughput rate, 30 °C water bath, 120 °C stretching chamber temperature, 50 s filament chamber heating time, and 1–2 or 5 s stretching time (depending on draw ratio). PET contents of 30, 40, and 45 wt% were realized using this optimization.

The stretching behavior of the neat polymers was studied with similar processing parameters. For neat PP, a stretching ratio in the range of 6–14 was possible. A lower stretching ratio led to inhomogeneous stretching, and filaments with a stretching ratio above 14 broke. For neat PET, a stretching ratio was only possible with a range of 5–6.

### 2.4. Thermal Characterization

#### Differential Scanning Calorimetry (DSC)

Differential scanning calorimetry tests were conducted for traces of stretched strands using a Mettler Toledo (Giessen, Germany), DSC821 device. Two heating runs were conducted. In the first heating run, the specimen was heated from room temperature to 200 °C and then cooled to 40 °C. In the second heating run, the specimen was reheated to 280 °C and kept at this temperature for 3 min before cooling down to 40 °C. The heating and cooling rate was kept constant at 20 °C/min. This test aimed to determine the melting ranges of the polymers to select the proper processing temperatures.

## 3. Results

### 3.1. Rheological Properties

Figure 2 shows the viscosity of PP and PET as a function of shear rate. The viscosity analysis was conducted at 275 °C. The objective of viscosity analysis is to investigate the extruding and the processability of the composite constituents and to determine the processing energy needed. As shown in the figure, PET is not a function of the shear rate. In contrast, polypropylene (PP) presents a pronounced dependency on the viscosity from the shear rate. PP behaves as non-Newtonian material while PET behaves as a Newtonian one for the frequency range tested.

### 3.2. Thermal Properties

Figure 3 shows different melting peaks for the PP/PET blend, from which the temperatures for extrusion mixing, filament stretching, and composite isotropization were derived. The selected extrusion temperature has to be above the melting peak of PET. A completely melted blend is necessary for good distribution quality [16,17]. The temperature should not be very high to avoid degradation of the PP. For further processing of the filaments, one has to realize a complete isotropization of the matrix (melted PP) without melting the fibrils.

### 3.3. Morphology

An important aspect for a successful fibrilization via the stretching of two mixed polymers is the appropriate distribution of the reinforcing phase (e.g., polyethylene terephthalate (PET)). The morphologies of the microfibrillar-reinforced composites blend were analyzed using a high-resolution (Carl ZeissGmbH, Oberkochen, Germany) scanning electron microscope. The extruded blend filament was immersed in a liquid nitrogen bath and then fractured. The specimen was gold-coated prior to conducting an SEM analysis, which was conducted for all ranges of the MFC blend. Figure 3 shows the morphology of the MFCs for 4 different weight percents of PET: 22, 30, 40, and 45 wt%. In all compositions, the SEM micrograph of the extruded polymer blends reveals a distinct two-phase isotropic morphology.

Figure 4a–c shows that the 22, 30, and 40 wt% PET is homogeneously dispersed in the PP matrix in the form of spherical particles or droplets with a diameter of 2–6 µm. Craters in the matrix of the same shape and size of particles were also observed. It can be inferred that the craters were created because of PET droplet pullout during cryogenic fracture. A PET content of 45 wt% led to large PET and PP spherical areas and smaller droplets inside this phase.

### 3.4. Optimization of Filament Stretching

During the fibrillation and orientation of the PET in the MFCs, the PP matrix also experienced stretching and consequently became highly oriented. Stretching (melt-spinning) of MFC constituents in a temperature below the melting point of the polymers resulted in very strong fibers with high modulus and strength [12,13]. The microfibril is the fundamental unit in an MFC structure. The microfibril block is composed of alternating layers, crystal, and amorphous layers. Amorphous layers consist of interconnected structural phases, mainly tie molecules, chain ends (cilia), and chain folds with varying loop length. Tie molecules connect the chain-folded blocks along the stretched direction of the MFC. Tie molecules enhance strength and modulus in the microfibril by connecting the blocks in the longitudinal (stretch) direction [14]. Because of the super-molecular structural organization of the two constituent phases, the stretched PP/PET blends possessed superior mechanical properties.

### 3.5. Isotropization

The isotropization of MFCs was carried out by heating the MFC to transform the PP matrix from a highly oriented to an isotropic matrix with the higher melting component’s fibrils of the PET. The MFC filaments were mechanically fixed and heated for 2, 4, and 6 min to three different temperatures, which were 168, 178, and 188 °C. Figure 5 shows the strength and the modulus versus the isotropization temperature.

### 3.6. Filament Optimization

The results of the tensile tests of selected filaments after isotropization treatment are shown in Figure 6. For an oriented filament prior to isotropization with 22 wt% PET processed with a stretching ratio of 15, very high modulus (8 GPa) and tensile strength (270 MPa) values were measured. The isotropization process relaxed the matrix and the modulus and strength dropped. The strength and modulus dropped as the holding time and temperature increased. The modulus dropped from 8. GPa for the highly oriented to 3 GPa for the isotropized filament at 188 °C for 6 min. Similar behavior of isotropization with the strength occurred where it dropped from 270 MPa to 120 MPa. The effect of the stretch ratio on the modulus of the neat polymers is shown in Figure 5. The neat PP and PET were stretched to their maximum stretching ratio. PP showed a continuous increase in the modulus with a higher stretching ratio, which led to an improvement of 550% (about 7 GPa) for a stretching rate of 15. PET filaments with a stretching factor of 6 possessed a modulus in the same range as a 15-fold stretched PP. Higher stretching of PET was not possible due to filament breakage.

The effect of parameter variations (filament speed, chamber temperature, and extruder melt temperature) on the mechanical properties are shown in Figure 7 and Figure 8. The mechanical properties of the stretched filaments depended strongly on the processing temperatures and speeds. At a lower melt temperature (*T_m_* = 255 °C), the modulus and strength were low and the increase in stretching ratio had no effect on their properties. At the melting temperature (*T_m_* = 269 °C), a linear increase of the modulus was observed with an increase in the stretching ratio (Figure 7). The modulus increased from 5 GPa at a stretching ratio of 9 to about 8 GPa at a stretching ratio of 14.

In Figure 8, strength reached a maximum at a stretching ratio of 11 and further increase in the stretching ratio resulted in a reduction in filament strength. Therefore, it was not possible to manufacture a filament with the highest modulus and strength. The best combination of tensile properties for the filament in terms of modulus/strength was produced at a stretching ratio of 12.6 (7.3 GPa/303 MPa) and 14.8 (8 GPa/270 MPa). From Figure 6 it can be summarized that all parameter variations showed a linear increase of the modulus with an increased stretching ratio. The dependencies of the tensile strength from the stretching ratio presented maxima at ratios of 10 or 12. An increase in melt temperature led to higher mechanical properties (increase > 60% at ratio 10 in Figure 6 and Figure 7). The highest modulus and strength were determined at a chamber temperature of 120 °C (stretching ratio 10, 12, in Figure 7 and Figure 8).

Very high stretching ratios (>12) showed a pronounced decrease in the strength, which could have been caused by inhomogeneous stretching (Figure 8). A reduction of the throughput rate showed a very high stretch ratio (stretching ratio 14.5) and it might be useful to investigate the potential of property improvement at lower stretch ratios. Furthermore, Figure 7 showed that for all parameter variations there was a linear increase in the modulus with an increased stretching ratio and maxima of the tensile strength at stretching ratios in the range of 10 to 12. An increase in melt temperature from 255 °C to 269 °C led to a pronounced increase in the modulus and strength (>60%) for 11-fold stretched filaments. A variation of the stretching chamber temperature led to an improvement of the modulus, which increased from 5.0 GPa to 6.4 GPa at 120 °C for a stretching ratio of 11. The dependency of strength on the parameter variation showed slightly higher values of strength with increasing chamber temperature and a maximum strength at 268 MPa for a stretching ratio of 11 (Figure 8). The maximum stretching ratios were found at a chamber temperature of 120°C, a melt temperature of 269 °C, and a reduction in the initial filament speed *v_1_* from 110 to 80 mm/sec before entering the chamber. This led to stretching ratios of 15.5 and produced very high modulus (8.3 GPa) and strength (280 MPa) values (Figure 7). The corresponding SEM micrograph of the blended morphology shows smaller PET droplets for a melt temperature of 269 °C (Figure 9). Further parameter optimizations were carried out in terms of various initial filament speeds (110, 80 and 57 mm/s) at a chamber temperature of 105 °C.

Further investigation on the effect of initial filament speed was conducted. Figure 10 and Figure 11 show the mechanical properties of the filament as a function of the stretching ratio at different initial speeds. Melt and chamber temperature were optimized in Figure 7 and Figure 8 as 105 °C and 269 °C, respectively. As shown in Figure 10, the young modulus reached the maximum value (8 GPa) at a stretching ratio of 14 at an initial speed of 85 mm/s. Stretching ratios in the range 12.6 to 14.8 present the maximum modulus of (8 GPa) and strength (303 MPa).

The morphology of the optimized MFC filament is investigated using an SEM micrograph. The MFC filament were immersed in liquid nitrogen bath and then fractured. Figure 12 shows the morphology of the MFC for PP/22 wt% PET. It is clear that PET fibrils have been developed in the PP matrix.

A comparison between the neat and MFC polymers is shown in Figure 13. For the optimized condition of speed, chamber temperature, and melt temperature, the effect of stretching ratio on modulus was investigated. MFC (PP/22 wt% PET) had the highest modulus value. Developing samples from the MFC filaments using extrusion or compression molding is the next subject that we plan to present, which will discuss mainly the MFC development and its characterization.

## 4. Conclusions

The development of polymer/polymer filaments that can be recycled and can replace fiber composites is the focus of this study. The development process was divided into two steps: (i) the development of MFC filaments was optimized with respect to good mechanical properties, a high stretching ratio, and a microstructure leading to outstanding mechanical and crash properties when further processed to composites. The structure–property relations (morphology versus mechanical) of MFC filaments were investigated according to the influence of anisotropy and arrangement of PET fibrils, aspect ratio, matrix morphology, and matrix/fibril adhesion. It can be summarized that the stretched filaments’ mechanical properties depend strongly on processing temperatures and speeds. A linear increase in the modulus was determined with an increased stretching ratio, and very high stretching ratios show decreased filament strengths. Therefore, it is not possible to manufacture a filament with the highest modulus and strength. The best combination of tensile properties in terms of modulus/strength show filaments produced at a stretching ratio of 12.6 (7.3 GPa/303 MPa) and 14.8 (8GPa/270MPa). The latter result came from the pullout of PET droplets during cryogenic fracture. A PET content of 45 wt% led to large PET and PP spherical areas and the inclusion of smaller droplets inside of these phases. An increased melt temperature from 255 °C to 269 °C led to a pronounced increase in the modulus and strength (>60%) for 11-fold stretched filaments. A variation of the stretching chamber temperature led to an improved modulus (for increasing temperatures) from 5.0 GPa to 6.4 GPa at a chamber temperature of 120 °C for this stretching ratio. The achievable maximum stretching ratios were found for a stretching temperature of 120 °C; melt temperature of 269 °C and a reduction of the initial filament speed *v_1_* before entering the chamber from 110 to 80 mm/s, which led to stretching ratios of 15.5 and caused very high modulus (8.3 GPa) and strength (280 MPa) values. Finally, it should be underlined that all proposed blends are entirely recyclable and can be re-used.

## Figures and Tables

**Figure 1 polymers-13-00233-f001:**
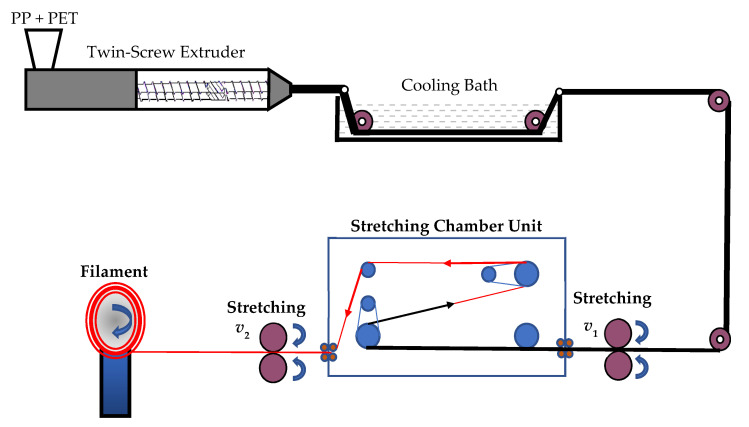
Schematic view of the MFC process.

**Figure 2 polymers-13-00233-f002:**
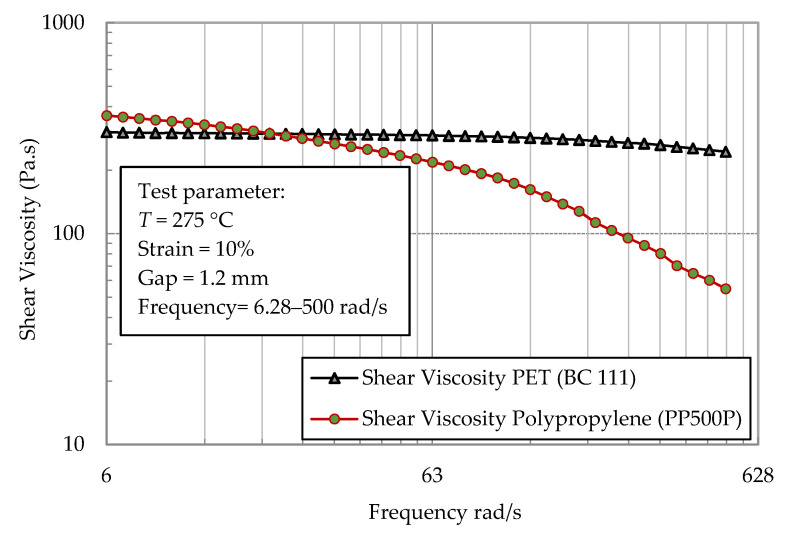
Rheology of PP and PET polymers.

**Figure 3 polymers-13-00233-f003:**
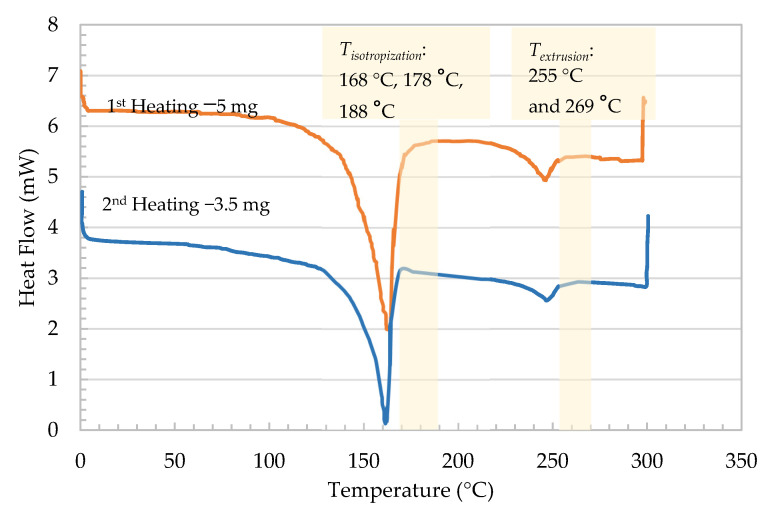
Melting behavior of PP/PET blends’ isotropization/processing temperatures.

**Figure 4 polymers-13-00233-f004:**
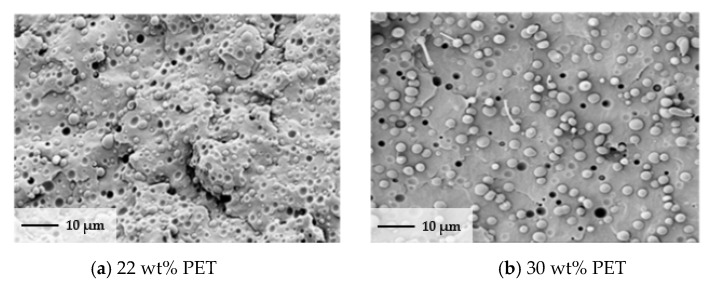
Morphology overview PP/PET of cryo-fractured SEM samples.

**Figure 5 polymers-13-00233-f005:**
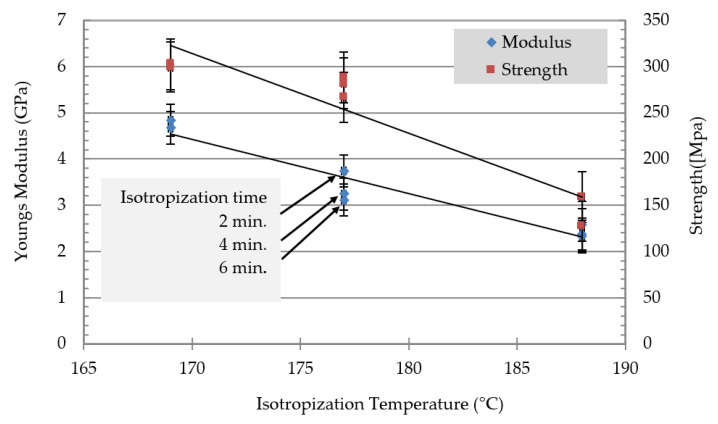
Isotropization of stretched MFC filament of 22 wt% PET.

**Figure 6 polymers-13-00233-f006:**
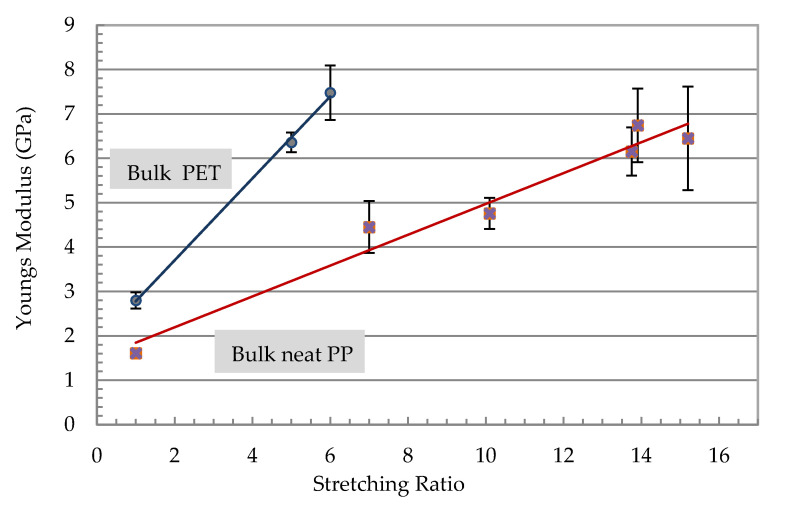
Young’s modulus of stretched neat polymer filaments.

**Figure 7 polymers-13-00233-f007:**
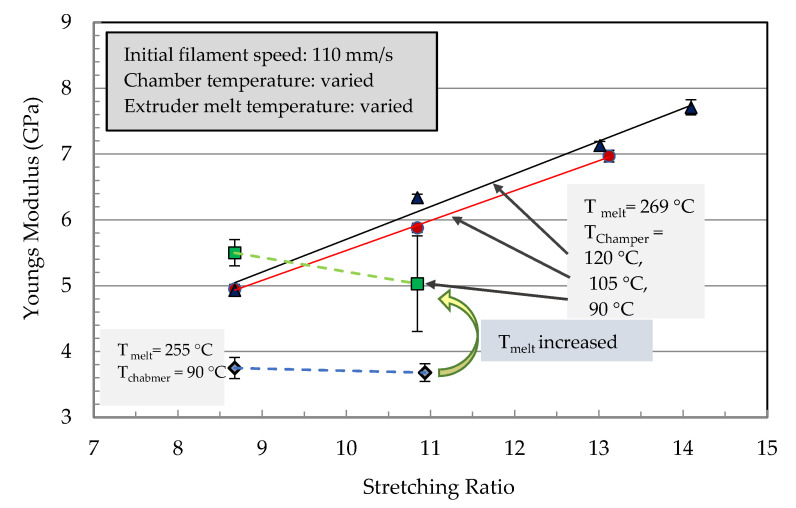
Young’s modulus of the stretched PP/22 wt% PET filaments as a function of the stretching ratio for different processing temperatures of the extruder (melt temperature) and the stretching chamber.

**Figure 8 polymers-13-00233-f008:**
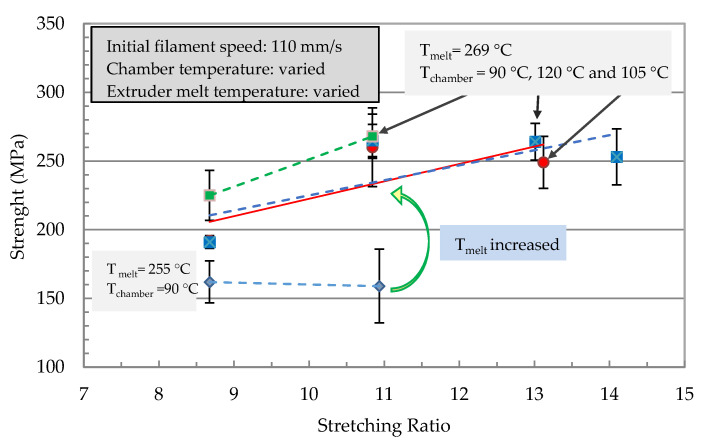
Tensile strength of the stretched PP with 22% PET filaments as a function of the stretching ratio for different processing temperatures of the extruder (melt temperature) and the stretching chamber.

**Figure 9 polymers-13-00233-f009:**
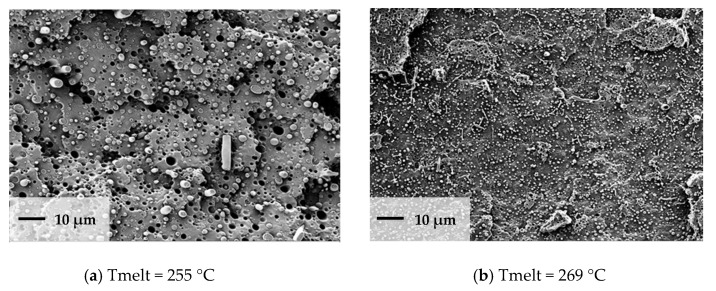
Morphology of PP 22 wt% PET blend: (**a**) melt temperature = 255 °C, (**b**) melt temperature = 269 °C.

**Figure 10 polymers-13-00233-f010:**
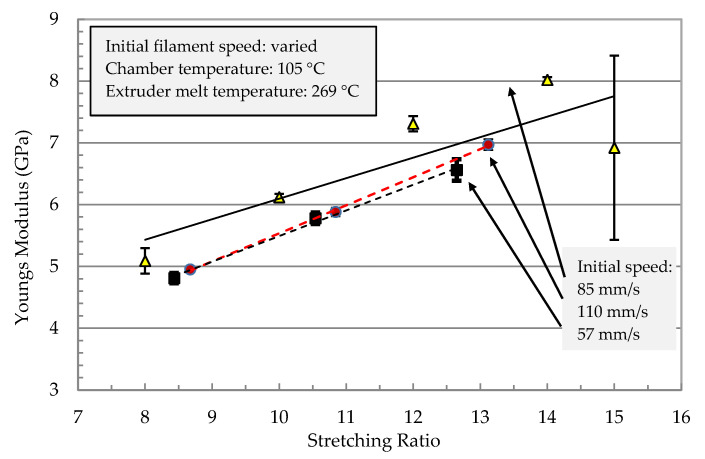
Young’s modulus of the stretched PP with 22 wt% PET filaments as a function of the stretching ratio for different initial filament speeds *v_1_* before entering the chamber.

**Figure 11 polymers-13-00233-f011:**
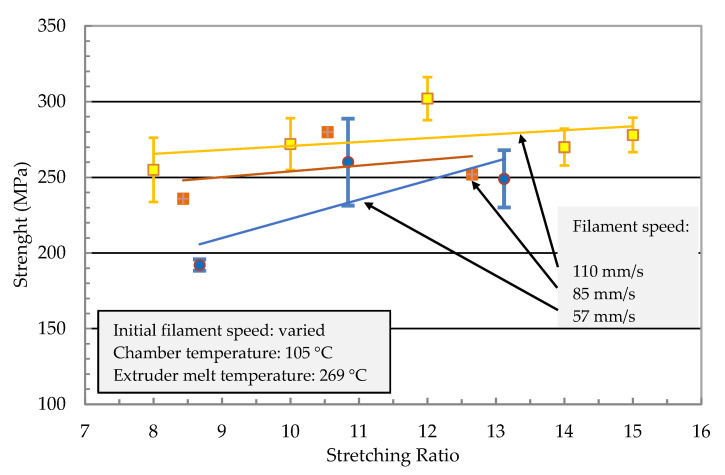
Strength of the stretched PP with 22 wt% PET filaments as a function of the stretching ratio for different initial filament speeds *v_1_* before entering the chamber.

**Figure 12 polymers-13-00233-f012:**
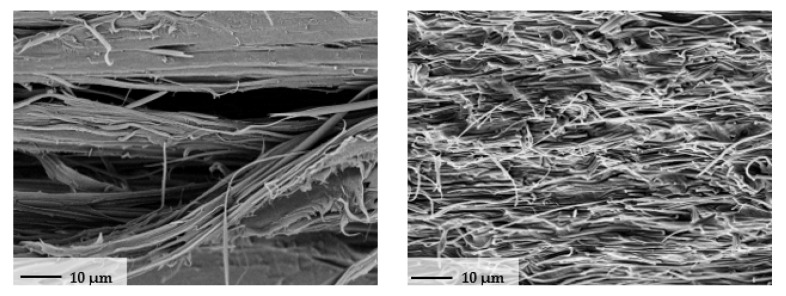
SEM micrographs of fractured PP/22 wt% PET.

**Figure 13 polymers-13-00233-f013:**
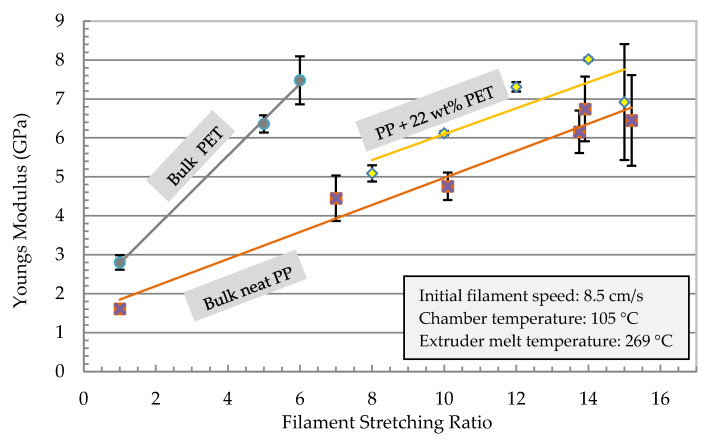
Young’s modulus of stretched neat PP, PET and PP/22 wt% PET filaments.

**Table 1 polymers-13-00233-t001:** Manufacturer’s datasheet of MFC-based polymers.

Polymer Type	Polymer Grade	Manufacturer	Density (g/cm^3^)	T_m_, proc. Temp. (°C)	MFR (dg/min)	Intrinsic Viscosity (dl/g)
PET	BC111	Sabic	0.838	246–256	-	0.74–0.78
PP	500P	Sabic	0.905	200–225	3	-

## Data Availability

The data presented in this study are available on request from the corresponding author.

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
