# Peer review of "Development of Polypropylene/Polyethylene Terephthalate Microfibrillar Composites Filament to Support Waste Management"

_polymers, 2021, doi:10.3390/polym13020233_

Round 1

Reviewer 1 Report

The effects of influence of the stretching ratio, extruder melt temperature, stretching chamber temperature, and filling speed on the properties of the PET/PP composites were discussed. In recent years, there have been a lot of research results on PET fiber modified PP. Therefore, the innovation of this paper is not strong. In addition, there are many low-level mistakes in the paper.

  1. Table 1. There is many error in the units.
  2. Figure 2. please use the original data to draw the graph, not the image generated by the instrument.
  3. Some specific experimental methods of analysis and test are missing in the experimental methods, please add them.
  4. Is the last paragraph of the introduction incomplete? Did the author upload the wrong version.
  5. What does the rheological behavior of materials explain? Is only PP dependent on shear rate? What's the meaning of this method for the following content?
  6. Why are the parameters of secondary melting peak and primary melting peak of blends so different. The author should give an account. The author should explain in detail why the conditions of primary melting are different from those of secondary melting.

Although the author has done a lot of work, I think this work lacks innovation and the author made a lot of low-level mistakes and should not be published.

Author Response

The author would like to thank the reviewer for valuable comment that help enhancing the quality of the paper. The response is attached.

Reviewer 2 Report

In this contribution by Almajid et al., PP and PET were used to form microfibrillar composites. The authors characterized the mechanical properties of these materials and studied how they vary with a change in PET content. The results show that the material has appreciable properties. The study is interesting and fits the scope of Polymers. However, some corrections have to be made to make this work publishable in this journal. Please see the suggestions below:
1) The authors are invited to improve the readability of this contribution as the flow of thought is very unclear. For instance, the abstract reads: "The two polymers were dry-mixed with PET contents ranging from 22 to 45wt%. The PET has been used as reinforcements to improve the mechanical properties of automotive components. The relationship
between the morphology of the MFC structure and the mechanical behavior of the MFC filament was investigated." The middle sentence appears out of context. The reviewer is aware of the potential application in the automotive industry, which was then highlighted in the introduction section, but this sentence is found at a very unexpected place. Please carefully go through the whole article to eliminate similar problems with narration.
2) There seems to be something missing in this part - maybe bulleting"
"Melt blending of two thermodynamically immiscible polymers with a difference in their melting temperatures of at least 30°K followed by
Cold stretching of the extruded blend above the glass transition temperature of both polymers to achieve axial orientation of the phases (fibrillization) and
Isotropization of the matrix polymer by a thermal annealing step between the melting temperatures of both components."
3) "In this work" section in the introduction part is non-existent: "In this paper, we present step (i): MFC development process." It is composed of only one short sentence. Please demonstrate to the readers the summary of what was done paying particular attention to describe the novelty factor well. What was done here that others did not do before? How these results fit the existing state of the art?
4) What was the MW of the polymers? Without such information, the study may be irreproducible.
5) "The granules were melted blended and extruded in a twin-screw extruder (PL2000, Brabender GmbH & Co., KG Duisburg, Germany) using standard processing parameters." - please specify what are the standard conditions. For instance, at present, the temperature is missing, which once again disables others to reproduce the study, which is one of the main goals of a research paper. In this case, others cannot build on this work, which very much limits its impact.
6) It would be useful to provide a scheme of the process.
7) Plots should have uniform formatting. Currently, almost every plot has a different theme, which makes the article very hard to follow.
8) Scale bar markers should have the same position (Fig. 3).
9) No error bars in Fig. 4. Please include them.
10) Larger magnification micrographs in Fig. 8 would help the readers to compare the samples.

Author Response

(The authors gave the same response as above.)

Round 2

Reviewer 1 Report

After the author's revision, I think it's ready for publication.

Reviewer 2 Report

The revisions completed satisfactorily. The article can be accepted for publication.